# Acetate Production from Syngas Produced from Lignocellulosic Biomass Materials along with Gaseous Fermentation of the Syngas: A Review

**DOI:** 10.3390/microorganisms11040995

**Published:** 2023-04-11

**Authors:** Budi Mandra Harahap, Birgitte K. Ahring

**Affiliations:** 1Bioproducts, Science, and Engineering Laboratory, Washington State University Tri-Cities, 2710, Crimson Way, Richland, WA 99354, USA; budi.harahap@wsu.edu; 2Department of Biological System Engineering, Washington State University, L. J. Smith Hall, Pullman, WA 99164, USA; 3Voiland School of Chemical Engineering and Bioengineering, Washington State University, Wegner Hall, Pullman, WA 99164, USA

**Keywords:** lignocellulose, acetogens, acetic acid, its application, syngas fermentation

## Abstract

Biotransformation of lignocellulose-derived synthetic gas (syngas) into acetic acid is a promising way of creating biochemicals from lignocellulosic waste materials. Acetic acid has a growing market with applications within food, plastics and for upgrading into a wide range of biofuels and bio-products. In this paper, we will review the microbial conversion of syngas to acetic acid. This will include the presentation of acetate-producing bacterial strains and their optimal fermentation conditions, such as pH, temperature, media composition, and syngas composition, to enhance acetate production. The influence of syngas impurities generated from lignocellulose gasification will further be covered along with the means to alleviate impurity problems through gas purification. The problem with mass transfer limitation of gaseous fermentation will further be discussed as well as ways to improve gas uptake during the fermentation.

## 1. Introduction

Acetate, an anion form of acetic acid, is an indispensable building block in a wide variety of applications, such as in the food sector, where acetate is transformed into sodium acetate for use as a food preservative [1] or into ethyl acetate as synthetic flavor enhancer [2]. Another application is within the chemical industry, where vinyl acetate monomer (VAM), a monomeric material of polyvinyl alcohol and polyvinyl acetate, is used for producing packaging materials, painting, and paper coating products. Adhesives and textile treatments also use up to 65% of the acetate produced worldwide [3,4,5,6]. Due to the broad application of VAM, the global market size of this platform chemical reached USD 8.97 billion in 2019, and this trend is projected to grow in the following years with a CAGR (compound annual growth rate) of 4.2% [7]. The increase of the VAM market or other acetate-derived chemical intermediates has driven high acetate demand (acetic acid), with its global market size reaching USD 9.30 billion in 2020 and a predicted CAGR of 5.2% [8]. Even though acetate is highly required for the chemical market, the current industrial acetate production still relies primarily on routes using unsustainable fossil-based fuels as raw materials [9].

Acetate commercially produced by chemical routes through oxidation processes uses various chemicals such as acetaldehyde, hydrocarbons, ethylene, ethane, methanol, and synthetic gas (syngas) generated from the petrochemical industries [10,11]. The drawbacks of this route, apart from the fact that non-renewable raw materials are used, are the need for toxic and corrosive catalysts, the high operational cost for catalyst preparation, the need for high pressure and temperature affecting the energy input needed, the demand for high substrate purity, a large amount of by-products being formed, and finally, high cost and complex processes of acetate purification and recovery. The biological route on the other hand offers the best option to cope with the serious problems of using the chemical route based on fossil fuels. The biological route can be operated under mild conditions with lower energy consumption and less need for purity, which fits with the use of renewable and sustainable raw materials such as biomass materials, agricultural commodities, organic waste, and synthesis gases [12].

One of the agricultural commodities and waste currently available is lignocellulosic-based material composed of C5 and C6 sugars, which can be fermented into acetate by heterotrophic acetogens. In general, bioconversion of lignocellulose into biochemicals, including acetate, involves pretreatment to overcome the recalcitrance caused by the presence of lignin and for increasing the enzymatic hydrolysis to produce hydrolysate sugars before the fermentation to transform monomeric sugars into acetate by heterotrophic acetogens [13] such as *Acetobacterium woodii* [14] and *Moorella thermoacetica* [15]. However, acetate production through this route is unable to optimally utilize all lignocellulosic components because this route only uses two of three major lignocellulose components, cellulose, and hemicellulose. Lignin is often used to produce steam or is treated as a final waste product [16]. Furthermore, the separation of lignin from sugar-rich hydrolysate can be difficult if the pretreatment develops a low-molecular-weight lignin compound [17]. Hence, other approaches using all the lignocellulose components for acetate production can be more beneficial. The discovery of autotrophic acetogenic bacteria that enable the utilization of CO and CO_2_ (syngas) and convert this gas to acetate has offered an alternative way of biological acetate production through integrated gasification and syngas fermentation processes [18]. In this integrated process scheme, lignocellulose biomass is thermally gasified to produce syngas composed of CO, CO_2_, and H_2_. In this process, the lignocellulosic components are used, and the products are valuable carbon and electron donors for acetogenic bacteria [19,20]. Direct synthesis of acetate from syngas using a metal catalyst such as V/TiO_2_ or salt catalyst of RuO_2_-CoBr_2_ and Bu_4_PBr are alternative routes to the biological route [21]. These processes, however, require high temperature and pressure up to 220 °C and 382 bar and will further produce by-products, mainly methyl acetate, ethyl acetate, water, and methane. Thus, converting syngas to acetate by fermentation using autotrophic and carboxydotrophic acetogens is a promising alternative.

Studies on syngas fermentation by solventogenic acetogens for bioethanol production have been widely reported and reviewed [22,23,24,25]. Several factors which play an essential role in enhancing the solventogenesis of syngas fermentation and the thermodynamic calculation of the Gibbs free energy of the alcohol production have also been discussed [26]. Furthermore, the review of the strategy to mitigate the energetic limitation of gases in syngas fermentation [27] and the ways to improve syngas fixation efficiency by genetic and metabolic engineering [28] have also been done. In addition to ethanol, several articles also have comprehensively reviewed the production of other products, such as medium-chain carboxylates [29] and methane [30], through syngas fermentation integrated with anaerobic fermentation. However, no review article has specifically discussed syngas-to-acetate fermentation and ways to improve acetate production, as done in the present review. Martin-Espejo et al. reviewed another sustainable route of acetic acid production from biogas-derived syngas through methanol carbonylation or non-thermal plasma that required the development of a robust catalyst and high energy requirement [31]. Thus, in this review article, we will focus on introducing mesophilic and thermophilic microorganisms that have the capability of syngas-to-acetate conversion, including their metabolic pathways. Furthermore, the optimization of fermentation conditions involving temperature, syngas composition, nutrients, syngas contaminants, and the strategy to enhance the mass transfer of syngas into the liquid phase is further elaborated in this review.

## 2. Acetate Production through Integration of Lignocellulose Gasification and Syngas Fermentation

Syngas production often uses renewable lignocellulose biomass materials such as agricultural and forestry residues besides municipal solid waste and energy crops [32]. The combination of thermochemical followed by biological conversion using lignocellulose material is a promising green technology for sustainable acetate production in the future. Figure 1 shows the integrated process of gasification and syngas fermentation to acetate. During the gasification, oxidizing agents such as air, oxygen (O_2_), steam, carbon dioxide (CO_2_), or hydrogen peroxide (H_2_O_2_) is added and used for partial combustion (auto-thermal process) or external heat (allo-thermal process) [33,34]. The oxidizing agents influence the final composition and the quality of syngas [35]. For example, steam gasification will produce a high H_2_/CO molar ratio, H_2_ concentration, and yield. On the other hand, elevated CO_2_ and CH_4_ concentration, lower heating value (LHV), and CO yield could be achieved in CO_2_ gasification. Using these conditions, the CO_2_ concentration and the gas production rate reached the highest-level during gasification. The use of H_2_O_2_ as an oxidizing agent of gasification could result in cleaner syngas due to lower char, oil, and water-soluble products [36].

The auto-thermal process of gasification is preferred over the allo-thermal process because this process can exothermically provide internal heat for the gasification process and allow energy recovery as shown in Figure 1. Lignocellulose gasification involving the auto-thermal process consists of four primary thermochemical stages [34]: 1. Drying to lower moisture and increase LHV, 2. pyrolysis to decompose dried biomass into syngas (H_2_, CO, CO_2_), CH_4_, tar, and char followed by 3. tar decomposition, oxidation of carbonaceous species for heat generation, and 4. reduction of products from pyrolysis and oxidation with the aim of producing CO, H_2_, and CH_4_. The output of the overall gasification consists of some valuable gases (syngas, CH_4_, other C_2_ and C_3_ hydrocarbon, and traces of other gases), tars in the form of unreacted liquid (heavy hydrocarbons and phenolic compounds), char as the unconverted solid residual (carbon and ash), and contaminants (nitrogenous, sulfurous species, and hydrogen halide) [37]. Tars, nitrogenous species (NO, NH_3_, and HCN), sulfurous species (H_2_S, CO_S_, CS_2_, and SO_2_), and Hydrogen halides (HCl and Cl) can potentially interfere with microbial growth and fermentation performances [38]. Thus, prior to the fermentation, the purification of crude syngas is necessary. The composition of crude syngas, with the contaminant present, is dependent on the type of gasifier and the feedstock selected [39].

After syngas purification, the selected autotrophic acetogens, mainly from the *Clostridium* genus [40], are grown to produce acetate via the Wood–Ljungdahl Pathway (WLP). Accumulation of acetate during the fermentation depends on the specific microorganism and the fermentation conditions [37] possessing different maximum acetate titer, yield, and productivity. Only a few studies have reported specifics of acetate production from syngas. Liakakou et al. did, for instance, find acetate productivity of 0.16 g/L/h using lignin and beech wood as raw materials and *Clostridium ljungdahlii* as the microbial catalyst [41].

As a result of the acetate production, acetate accumulated in the fermentation broth needs to be separated and further purified before it can be marketed. The fermentation broth before separation and purification contains remaining media components such as cells, potential alcohols and/or other organic acids besides acetate [42,43]. Several acetate separations and purification techniques have been described involving adsorption, solvent extraction, precipitation, distillation, reactive distillation, and membrane processes [11].

## 3. Microbiology of Syngas-to-Acetate Fermentation

### 3.1. Microorganism

Syngas fermentation for the production of biochemicals or biofuels predominantly involves autotrophic acetogens using the Wood–Ljungdahl pathway (WLP), also known as the reductive acetyl-coenzyme A (Acetyl-CoA) pathway [44]. Acetogens use C1 gases (CO and/or CO_2_) as electron acceptors, and CO and/or H_2_ as the electron donor for the reduction of CO_2_ into acetyl-CoA as an intermediate product for producing alcohols and various organic acids, including acetate. The acetate yield obtained during this fermentation is highly dependent on the type of microorganism used [40]. Thus, selecting the right microorganism is essential for obtaining high acetate yield and productivity.

#### 3.1.1. Grow in Axenic Culture

The syngas fermentation by axenic culture can create reproducible data [45], and the process is also more easily controllable than mixed culture because a single type of microorganism in the axenic culture has consistent nutrient requirements, growth rate and other physiological characteristics of the cell [46]. Laboratory-scale research on syngas-to-acetate fermentation has, therefore, often used pure cultures. The fermentation by pure culture could use mesophilic or thermophilic bacteria. The classification of the different acetogens, along with their optimum growth conditions and by-products produced besides acetate, are shown in Table 1.

Compared to thermophilic autotrophic bacteria, more mesophilic autotrophic bacteria have been isolated and further studied for their performance during the fermentation of C1 gases (CO and CO_2_) into various biochemical products (Table 1). The advantages of mesophilic acetogens for syngas-to-acetate fermentation over thermophilic acetogens are lower energy consumption for process heating and more genetic tools available for the genetic engineering of these strains. Most mesophilic acetogens described can both live autotrophically using inorganic compounds such as CO_2_ or CO as the carbon source or live heterotrophically on organic compounds such as *Clostridium ljungdahlii*, *C. carboxidivorans*, *C. ragsdalei*, *C. autoethanogenum*, *C. drakei*, *Alkalibaculum bacchii*, *Butyribacterium methylotrophicum*, and *Oxobacter pfennigii*. The products produced are either ethanol, butyrate, butanol, lactate, or 2,3-butanediol apart from acetate. For instance, *C. carboxidivorans*, shown in Figure 2, accumulated butyrate, butanol, and ethanol and then released these products outside the cell. The stoichiometric reaction and Gibbs free energy related to the production of these products can be seen in Table 2. As a consequence of the production of other products besides acetate, the yield of acetate will decline.

A batch fermentation test of *C. carboxidivorans* using 2.2 atm CO gas maximally yielded 1.87 g/L acetate and formed several by-products such as butyrate (0.23 g/L), caproate (0.09 g/L), ethanol (0.19 g/L) and butanol (0.075 g/L) [69]. Meanwhile, batch fermentation by *C. ljungdahlii* maximally generated 2 g/L acetate using 2.0 H_2_/CO ratio, 0.35 g/L ethanol, and 0.5 g/L 2,3-butanediol using 0.5 H_2_/CO ratio [70]. The other mesophilic syngas bacteria, such as *C. ragsdalei*, *C. autoethanogenum*, *C. drakei*, *A. bacchii*, *B. methylotrophicum*, and *O. pfennigii* maximally produced acetate up to 12.3 g/L [71], 5.33 g/L [72], 0.5 g/L [73], 1.2 g/L [74], 1.3 g/L [75], and 0.16 g/L [76], respectively.

On the other hand, several mesophilic bacteria are categorized as autotrophic homoacetogens, such as *C. aceticum*, *A. woodii*, *P. productus*, and *E. limosum*, solely producing acetate with a maximum concentration of 7.2 g/L [77], 3.17 g/L [78], 3.2 g/L [79], and 0.5 g/L [80], respectively. These strains are, therefore, of special interest for large-scale acetate production.

In addition to acetogenic bacteria, methanogenic archaea have also been investigated for acetate production besides methane, formate, and methylated thiols from C1 gases. *Methanosarcina acetivorans* is, until now, the only acetate-producing archaea successfully identified, and the concentration of acetate produced was low (0.06 g/L) [81]. Rother [66] reported that acetate and formate are the primary metabolites of these archaea over methane. Acetate might, therefore, be produced by the multienzyme complex of CO dehydrogenase (CODH)/acetyl coenzyme A synthase (ACS), converting CO and CO_2_ into acetyl-CoA, an intermediate product for acetate formation and other metabolites [64]. The CO dissolved in the fermentation broth is oxidized to form CO_2_, and then the electron released from this oxidation step is used for CO_2_-to-a methyl group reduction and ultimately towards acetate synthesis [65].

Thermophilic microorganisms for syngas fermentation have recently attracted increased attention due to the higher growth rate of thermophiles compared to mesophiles. Furthermore, higher temperatures minimize the risk of contamination [82]. The thermophilic bacteria that have been reported to have the capability of syngas-to-acetate production include *M. thermoacetica*, previously known as *Clostridium thermoaceticum*, as well as *Desulfotomaculum kuznetsovii*.

The thermophilic homoacetogen of *M. thermoacetica* is classified as a gram-positive bacterium that can utilize both CO and CO_2_ to produce acetate [83,84]. Furthermore, unlike most mesophilic microorganisms that generate higher alcohols, acetate is the only product from syngas fermentation by *M. thermoacetica* [85]. This thermophilic bacterium was able to grow in various syngas blend compositions such as H_2_/CO_2_, 100% CO, and CO/CO_2_ fermentation [86], and the results of all these experiments showed that acetate could be produced (Table 3). For instance, Kerby and Zeikus reported that *C. thermoacetium* deposited by Fontaine et al. (currently named *M. thermoacetica* DSM 2955^T^ or ATCC 35608) rapidly grew on H_2_/CO_2_ and needed adaptation to grow on CO [87]. This bacterium strain stoichiometrically required 4.44 mol H_2_ and 2.33 mol CO_2_ to produce 1 mol acetate or 3.64 mol CO to produce 1 mol acetate. Another strain (*M. thermoacetica* DSM 521^T^) can further oxidize CO to CO_2,_ followed by CO_2_ reduction to acetate [49,88]. A mutant strain of *M. thermoacetica* (currently deposited as *M. thermoacetica* ATCC 49707 and DSM 6867) derived from *Clostridium thermoaceticum* ATCC 39289 has been described [89]. This *Moorella thermoacetica* strain generated up to 31 g/L acetic acid on 70% CO/30% CO_2_ at pH 6.0 in a bubble column bioreactor, a higher concentration as found with 40% CO/30% H_2_/30% CO_2_ fermentation of 26 g/L [90]. Genetic modification of *M. thermoacetica* is eased by the fact that the genome of this strain has been fully sequenced, as reported for *M. thermoacetica* DSM 521^T^ [91] and DSM 2955^T^ [91]. Hence, this bacterium is a good candidate for acetate production from syngas.

Another species of *Moorella* formerly named *Clostridium thermoautotrophicum* JW701/3 (*M. thermoautotrophica* ATCC 33924 and DSM 1974) isolated by Wiegel et al. could utilize CO_2_/H_2_ or CO for cell and acetate production [92]. Furthermore, Savage et al. reported that faster CO-dependent growth of this species of *Moorella* was reached by supplementing CO_2_ into the headspace [86]. However, *M. thermoautotrophica* is currently proposed to be on the list of rejected names because, based on phylogenetic and genomic analysis, *M. thermoautotrophica* and *M. thermoacetica* are closely related and show genetic similarities at the species level [93,94].

**Table 3 microorganisms-11-00995-t003:** Comparison of acetate production by several strains of *M thermoacetica*, their syngas fermentation conditions, and bioreactor type.

Strain	Syngas Type and Composition [-]	Additional Carbon Source [g/L]	Temp. [°C]	pH [-]	Bioreactor Type [-]	Acetate Production	Ref.
DSM 2955 or ATCC 35608	20% CO_2_/80% H_2_ 100% CO	-	55	7	Bottle	4.72 g/L 4.49 g/L	[87]
DSM 521	100% CO_2_ 20% CO_2_/80% H_2_ 100% CO	18 g/L glucose - 20 g/L glucose	55	7	Bottle	NA	[49,88]
DSM 6867 or ATCC 49707	40% CO/30% H_2_/30% CO_2_ 70% CO/30% CO_2_	-	60	6	Bubble column bioreactor	26 g/L 31 g/L	[90]
DSM 1974 or ATCC 33924	20% CO_2_/80% H_2_	-	56–60	5.7	Bottle	3.78 g/L	[92]
ATCC 39073	0.4 L/min CO_2_	10 g/L glucose	60	6.5	Stirred tank bioreactor with continuous syngas flow	9.3 g/L	[95]
JCM 9320	100% CO_2_	-	55	7.0	Microbial electrosynthesis	0.024 g/L/day	[96]

Other potential thermophilic bacteria are *D. kuznetsovii* and *D. thermobenzoicum* subsp. *thermosyntrophicum*, capable of growing autotrophically on mixtures of gases of H_2_/CO_2_ [97,98]. Both bacteria can further reduce sulfate (SO_4_^−^)/sulfite (SO_3_) while utilizing CO and H_2_ as the electron donor to generate H_2_S [50]. Moreover, besides H_2_S, acetate will be produced up to 0.4 g/L for *D. thermobenzoicum* subsp. *thermosyntrophicum* and 0.44 g/L for *D. kuznetsovii* using CO. Hence, the use of this microorganism brings about two functions: acetate production as well as the removal of SO_4_^−^.

Archaea such as *Archaeoglobus fulgidus* have also been reported to use CO, apart from CO_2_, as the carbon source for growth [99]. The presence of CO led to accumulation of CO_2_, acetate (18 mM), and formate (8.2 mM), while no H_2_ was detected. Another study also revealed that strains of this archaea genus effectively reduce SO_4_^−^ with CO as an electron donor [100] but have less need for the sulfurous compound for growth [99].

#### 3.1.2. Mixed Culture

Anaerobic fermentation by mixed cultures has further shown to be effective in producing acetate with high yields. Singla [101] mentioned that using mixed cultures has several benefits, such as no need for sterilization, more adaptive microorganisms that works synergistically together due to the microbial diversity along with the capability to utilize mixed substrates with lower operation costs, and long-term stability of growth. In addition, mixed cultures were found to tolerate higher variations in the syngas composition compared to pure cultures, where consistency often is a must [46]. Furthermore, acetate yields of up to 42 g/L have been reported by Wang [102] during mixed culture fermentation.

Syngas fermentation for acetate production by mixed cultures has been tested with both mesophilic and thermophilic inocula (Table 4) [4]. Wang reported that syngas fermentation with an inoculum containing thermophilic bacteria dominated by the genera *Thermoanaerobacterium* and *Thermohydrogenium* effectively produced acetate in both batch and continuous culture [102]. Similar studies were done with a mesophilic inoculum dominated by strains belonging to the *Clostridium* genus at temperatures from 35 to 37 °C [60,103,104,105]. Even though methanogens generally outcompete acetogens on CO_2_ and H_2_ under both mesophilic and thermophilic conditions, adjustment of the fermentation temperature and/or pH, or the addition of 2-Bromoethane sulfonic acid (BES) during start-up can inhibit the activity of methanogens and make acetogens prevail over methanogens [106].

### 3.2. C1 Metabolism by Acetogens

Homoacetogens are acetogens who can live solely metabolizing syngas (CO, CO_2_, and H_2_) through two branches of the Wood–Ljungdahl Pathway: the eastern or western branch to form acetyl-CoA as an intermediate ending as acetate (Figure 2). Furthermore, other products such as volatile fatty acids (butyrate and hexanoate), alcohol (ethanol, 2,3-butanediol, butanol, hexanol), and lactate can be produced. In general, one molecule of CO_2_ is reduced by six electrons with 1 ATP to form methyl tetrahydrofolate (CH_3_-THF) in the eastern branch, whereas the C1 substrate is directly converted into acetyl-CoA using 2 electrons in the western branch [107]. The acetyl-CoA formed is then converted to acetate while releasing ATP by substrate-level phosphorylation (SLP). The electron used during Wood–Ljungdahl Pathway comes from CO and H_2_ oxidation, whereas CO_2_ and CO serve as carbon sources.

#### 3.2.1. CO Metabolism

Carbon monoxide (CO), a major gas generated from gasification, is generally a toxic gas for most microorganisms because this gas has a high affinity for metal-containing enzymes, causing the inactivation of these enzymes. However, some bacteria have high resistance to CO and are named carboxydotrophic acetogens (see Table 1). These microbes are capable of CO oxidation to generate electrons and provide reducing equivalents catalyzed by carbon monoxide dehydrogenase (CODH) while transferring the formed electron to Ferredoxin (Fd) or other electron acceptors [108].

In carboxydotrophic acetogen metabolism, CO can act as both a carbon source and an electron source. Meanwhile, H_2_ only serves as an electron donor and requires a carbon source obtained from CO and/or CO_2_ [109]. The electrons or reducing equivalents are generated from biological water gas shift reaction (BWGSR) of CO (see Equation (1)) or hydrogen oxidation (see Equation (2)) that will further be used for CO_2_ reduction through the eastern and western branches of Wood–Ljungdahl Pathway [67]. When fermentation is performed in the absence of H_2_, only half of the carbon in CO is transformed into acetic acid, whereas the remaining carbon of CO is converted into CO_2_ (see Equation (3)). On the other hand, when H_2_ is available in the fermentation media, two mol CO_2_ is required to form one mol acetic acid [110] (see Equation (4)). Such CO_2_ is obtained from CO oxidation or supplying CO_2_ gas into the fermentation system.
CO + H_2_O → CO_2_ + 2H^+^ + 2e^−^(1)
H_2_ → 2H^+^ + 2e^−^(2)
4 CO + 2 H_2_O → CH_3_COOH + 2CO_2_(3)
2 CO_2_ + 4 H_2_ → CH_3_COOH + H_2_O (4)

CO-to-acetic acid conversion through Wood–Ljungdahl Pathway is catalyzed by oxidoreductase enzymes. For instance, the enzyme that plays an essential role in CO metabolism to acetate is carbon monoxide dehydrogenase (CODH). According to its role in the metabolic pathway, CODH is classified into a monofunctional CODH and a bifunctional CODH/acetyl CoA synthase (ACS). Monofunctional CODH plays a role in catalyzing reversible CO-to-CO_2_ oxidation, while bifunctional CODH, together with ACS, is involved in the reaction of CO, methyl-CoFeSP, and CoASH to form acetyl CoA [111,112]. The methyl-CoFeSP is formed by transferring the methyl group generated from the eastern branch of Wood–Ljungdahl Pathway to corrinoid/iron-sulfur protein (CoFeSP) catalyzed by a methyltransferase (MeTr); the mechanism can be seen in Figure 2. The catalytic reactions of both monofunctional CODH and bifunctional CODH/ACS are shown in Equations (11) and (12) of Table 5, respectively.

The enzymatic reaction mechanism of CODH was proposed by Can [113] as a ping-pong reaction. The metal center (Ni for most of the acetogens including *M. thermoacetica*) of CODH C_red1_ in the C-cluster bind CO by releasing one H^+^ and forming C_red1_-CO_2_, then reacting with H_2_O by releasing another H^+^ and CO_2_ to form C_red2_. The electron produced is then transferred to ferredoxin to re-form C_red1_, as shown in Figure 3.

Some studies have successfully purified and characterized both enzymes from anaerobic carboxydotrophic acetogens and attempted to describe the structure [114]. The CODH in anaerobic acetogens is structured by a complex metal center consisting of nickel (Ni), iron (Fe), and sulfur (S), serving as CO and H_2_O reaction center, intermediary metal-carboxylate stabilization, and temporary storage of two electrons from the reaction [115]. On the other hand, aerobic carboxydotrophic bacteria employ Cu, Mo, and Fe-containing flavoenzymes [112]. Doukov [116] described the biofunctional CODH/ACS. In *M. thermoacetica*, this is composed of iron (Fe), sulfur (S), copper (Co), and nickel (Ni) metallocofactor. Another study mentioned that the active site of both enzymes, the C-cluster and A-cluster for CODH and ACS, respectively, has a [Fe_4_S_4_] cluster connected to Ni [114].

#### 3.2.2. CO_2_ Metabolism

Chemolithoautotrophic acetogens utilize inorganic carbon (CO_2_) as the carbon source for product and cell formation through the eastern and western branches of WLP [44]. In the western branch, the conversion of CO_2_ into acetate involves several enzymes and proteins that are successfully identified: (1) CODH, (2) CODH/ACS, (3) corrinoid iron-sulfur protein (CoFeSP), two subunits, (4) MeTr, whereas the enzymes in eastern branch involve: (1) FDH, (2) FTS, (3) MTC, (4) MTD, and (5) MTR [111,112]. According to Figure 2, when acetogens grow with CO_2_ and H_2_ without CO in the fermentation media, CODH catalyzes the reduction of CO_2_ into CO in the western or carbonyl branch using the reducing equivalents produced from hydrogen oxidation. Meanwhile, CO_2_ is further reduced to form formate in the eastern branch. The formate formed in the last reaction is a precursor of the methyl groups of acetyl Co-A.

In the eastern branch, the formate formed from a two-electron reduction of CO_2_ is condensed with THF, involving ATP to form formyl-THF [44]. The formed formyl-THF subsequently undergoes cyclization to methenyl-THF catalyzed by MTC. Consecutive reduction then occurs of methenyl-THF to form methylene-THF, followed by methylene-THF to form methyl-THF. The CH_3_-THF is then transferred to the cobalt center of CoFeSP for CH_3_-CoFeSP formation. Together with CO, CH_3_-CoFeSP undergoes condensation to produce acetyl-CoA, then further reaction to form acetate. Finally, the formation of acetate is catalyzed by two enzymes, PTA and ACK, by generation ATP [47]. The detailed metabolic pathway and biochemical reaction of CO_2_ can be seen in Figure 2 and Table 5, respectively.

## 4. Optimization of Syngas Fermentation Condition

### 4.1. pH and Temperature

Controlling pH, apart from influencing cell growth, could regulate the metabolic pathway toward accumulating one specific product from homoacetogenic bacteria. Otherwise, more undesirable metabolites could possibly be produced, which would decrease the expected yield of the specific product of interest. For acetate production in particular, the fermentation generally must be maintained at neutral pH to obtain significant concentrations of acetate. The studies conducted by Abubackar et al., Kundiyana et al., Infantes et al., and Arslan et al. [22,117,118,119] using *C. autoethanogenum*, *C. ragsdalei*, *C. ljungdahlii*, and *C. aceticum*, respectively, reported that a pH drop from neutral to acidic pH could switch the process from acidogenesis to solventogenesis during anaerobic syngas fermentation. This process switch decreased the acetate yield and led to higher concentrations of ethanol formed [120].

The acetogens optimally grow and produce high acetate concentrations at controlled neutral pH during pure culture fermentation. When acetic acid is released into the medium, the internal pH of the cell decreases due to more passive H^+^ diffusion inside the cell through the cytoplasmic membrane; acetic acid is categorized as lipophilic acid with high-affinity towards cell membrane [121]. Adjusting the external pH to neutral conditions could mitigate the unfavorable internal pH, so the acetogens could normally grow and metabolize C1 gases to acetate [117,118,120]. Overall, lowering the external pH leads to stress conditions for the microbial cell inhibiting growth. The cell subsequently directs the metabolism towards alcohol formation to counteract this situation. In addition to cell metabolism redirection, the change in the pH also highly affects the CO dehydrogenase (CODH) performance, the key enzyme catalyzing acetyl-CoA formation and influencing acetate synthesis [52]. After RT-qPCR analysis, the lower CODH gene copy number was detected in the cells when grown at pH below and above pH 7, corresponding to a decrease in acetate formation. In addition, the acetogens reached the highest cell growth rate at pH 7. Because acetate is a primary metabolite related to growth, the highest acetate production is at that pH.

The pH adjustment strategy of mixed culture fermentation is slightly different from pure culture fermentation. For instance, a mixed culture inoculum obtained from anaerobic reactors operated on municipal wastewater was composed of a diverse community of microorganisms, including methanogens [103]. During the fermentation, the uncontrolled methanogenic population triggered the increase of syngas-to-methane and acetate-to-methane conversion by hydrogenotrophic and acetotrophic methanogens, respectively, that will lower the acetate yield and result in higher pH and methane production [44]. As with the acetogens growing on syngas, the methanogens reached the maximum numbers at neutral pH. Thus, it is important to eliminate methanogens in the inoculum either by heat, high or low pH, or the addition of 2-bromoethanesulfonate (BES) before using the mixed culture for acetate production. The heat-shock treatment can eliminate non-spore-forming bacteria such as hydrogenotrophic methanogens while spore-forming bacteria, like most species of *Clostridium* can grow due to protective spore formation [122]. In general, methanogens optimally grow around pH 7.0, while pH between 5.5 and 6 is a favorable condition for acetogens to produce high acetate concentration and yield without methane production [123]. Adjustment to pH 8–10 also can suppress methanogen population growth, but the alkaline condition can lower the rate of volatile solid degradation to acetate and lead to corrosion risk for the equipment [124]. Meanwhile, the addition of BES for a short period only clears out the methanogens and still allows the acetogens to grow as BES has a structural analog of Coenzyme-M that can block methyl Coenzyme-M (Me-CoM) formation pathway as a competitive inhibitor [125,126].

Higher acetate production with minimal ethanol and methane production was found by Liu et al. [103] using a pH of 9.0 and a mixed culture inoculum dominated with the genera *Clostridium* and *Acetobacterium*. Luo [105] further reported that acetate production reached the highest yield of a mixed culture with low concentrations of methane, butyrate, and alcohol at pH 6.5 and 7.5 with BES added during the fermentation. Acetogens were metabolically more active at pH 7.0, as seen by a higher cell number and species diversity [52]. However, BES is a costly substance to add during fermentation and should be limited to periodic additions in continuous reactors.

In addition to pH, the fermentation temperature has an essential role in the regulation of acetate production. By adjusting the temperature of the fermentation, the amount of acetate could change. For example, Shen [127] revealed that the fermentation temperature change from 37 to 25 °C caused a decrease in acetate concentration due to organic acid re-assimilation by *C. carboxidivorans*. Another study mentioned that the acetate concentration increased with a temperature increase from 32 to 37 °C for *C. ragsdalei*, while a higher temperature up to 42 °C resulted in a lower acetate concentration [117]. Thus, optimization of fermentation temperature is required. The optimum temperature of various acetogens can be seen in Table 1.

Fermentation at a temperature above 50 °C is preferred because this condition could reduce the energy requirement for syngas cooling [99]. However, the use of high temperatures during syngas fermentation is challenged by the fact that the mass transfer rate of syngas into the fermentation broth is lower at higher temperatures due to a decrease of syngas solubility in the fermentation broth with increasing the temperature [128]. When the amount of syngas dissolved in the fermentation broth is limited, cell growth and acetate formation will decrease.

During mixed culture fermentations, the temperature could be used to suppress the methanogenic population. Most acetogens grow at low temperatures, whereas methanogens have a high reduction in the growth rate when the fermentation temperature drops [74]. Hence, adjusting the temperature from mesophilic to psychrophilic conditions at 20℃ will inhibit the methanogens, while the acetogens will still grow and produce acetate [103]. However, the acetate production was 9% lower, and the conversion rate was also lower than to mesophilic condition.

### 4.2. Partial Pressure and Syngas Composition

Each gas in the syngas blend has different physiochemical and thermodynamic properties, affecting the gas transfer rate into fermentation media, the cellular gas uptake rate, and the cellular reaction rate. The gas physiochemical properties involve Henry’s law constant or solubilities. For example, CO_2_ has the highest solubility in water, shown with the highest Henry’s law constant of CO_2_ (K_H_ = 3.4 × 10^−2^ M/atm) over CO (K_H_ = 7.4 × 10^−3^ M/atm) and H_2_ (K_H_ = 7.8 × 10^−4^ M/atm) [129]. In addition, Geinitz et al. also mentioned that CO and H_2_ was far less soluble in water compared to CO_2_ shown by the maximum solubility in water at 303.15 K and 1 atm: CO (0.92 mmol/L), H_2_ (0.76 mmol/L), and CO_2_ (29.99 mmol/L).

On the other hand, in thermodynamic aspects, Hu et al. reported that electron generated from CO was more thermodynamically favorable than electron production from H_2_ with independence of pH, ionic strength, gas partial pressure, and electron carrier pairs [130]. This result indicated that CO was the preferred electron source rather than H_2_. Hence, the bacteria produced more electrons from CO with higher CO oxidation rate as compared to the electron production from H_2_. However, physiochemical and thermodynamic properties are not the only factor influencing gas transfer rate into fermentation media, cellular uptake rate, and cellular reaction rate.

According to the Henry’s law equation, the partial pressure of each gas proportionally contributes to the gas amount in the liquid phase apart from the Henry’s law constant. By having a high partial pressure in the fermentation system, the mass transfer limitation is expected to be mitigated, and more gas is ultimately dissolved into the liquid phase. The vast amount of syngas in the liquid phase can favor autotrophic acetogen growth to produce more acetate. Hurst et al. reported that increasing the CO partial pressure from 0.35 to 1.05 atm in a sole CO fermentation could produce more biomass of *C. carboxidivorans* P7 as well as acetate [109]. However, when the CO pressure was elevated above 1.05 atm, the acetate decreased due to more ethanol accumulation, while cell production continued up to a CO pressure of 2.0 atm. Above this pressure, cell production was inhibited by CO. Lanzillo et al. also revealed that CO partial pressure was responsible for the kinetic growth of *C. carboxidivorans* P7 with the optimal P_CO_ of 1.1 atm [69]. The cell growth rate increased because high CO partial pressure accelerated the CO transfer rate into the liquid phase. Nevertheless, this study also showed that too high CO concentration in the liquid phase could further decrease cell growth rate due to CO inhibition.

In a similar work using a sole CO substrate with different bacteria, *C. ljungdahlii*, reported that pressurizing the system up to 4 bars during stirred batch fermentation enhanced the CO solubility, the conversion of CO (98.5%), the growth phase, the cell numbers and acetate production with lower concentrations of ethanol generated [131]. The effect of increased partial pressure was further tested in a cell-recycled continuous bubble column bioreactor using *E. limosum* [132]. The maximum growth rate (q_CO_X) was reached at 74 kPa. Cell concentration decreased below that pressure due to a low mass transfer rate (K_L_a [C* − C_L_]) issue, whereas above 74 kPa of CO partial pressure, the maintenance requirement (m_s_X) was increased.

The partial pressure of each gas in the syngas mixture is also associated with the syngas composition and the concentration of each gas. In lignocellulosic material gasification, syngas composition and concentration vary depending on biomass properties (size, shape, density, chemical composition, energy, moisture content, reactivity, ash content, and volatile compounds), the type and design of gasifier, and the operational condition (temperature, pressure, oxidizing agent type, and flow, biomass flow and type, and catalyst amount) [133]. The CO_2_ in the lignocellulose-derived syngas mixture typically ranges between 5 and 15%, whereas the H_2_/CO ratio is at the interval of 0.2–4.6 [134]. The ratio of each gas in the syngas blend influenced the product composition during syngas fermentation when using solventogenic autotrophic acetogens [70]. For example, the CO concentration available in excessive amounts could promote the formation of more electron-dense products such as ethanol, over the less electron-dense product, acetate, during fermentation [70]. According to Wood–Ljungdahl Pathway, the CODH enzyme catalytically oxidized CO into CO_2_ by releasing reducing equivalents that were further utilized for Acetyl Co-A reduction into ethanol. The more CO is present, the more reducing equivalents are available for ethanol production. Furthermore, thermodynamical aspects also show that the CO conversion into ethanol through biological water-gas shift reaction (BWGSR) is thermodynamically more favorable due to its high free Gibbs energy released (ΔG° = −217.4 kJ/mol) over production of acetate (ΔG° = −154.6 kJ/mol) (See Table 2). Thus, when using solventogenic acetogens for acetate formation it is important to control the syngas composition of electron donor and acceptor.

One of the strategies to control more ethanol accumulation than acetate is by adjusting dissolved CO during fermentation by solventogenic autotrophic acetogens. For instance, a study conducted by Jack et al. and Esquivel-Elizondo et al. using either *C. ljungdahlii* or a mixed culture, respectively, reported that the use of a higher H_2_/CO ratio could accumulate more acetate over ethanol [46,70]. In this work, fewer reducing equivalents were formed from CO; instead, H_2_ provides the reducing equivalents for further use as CO_2_ reduction into CO and methyl-CoFeSP to produce acetyl CoA followed by acetate formation.

Excessive amount of reducing equivalents from H_2_, however, should also be avoided since more reducing equivalents caused more ethanol formation. For example, the study conducted by Valgepea and Mann using *C. autoethanogenum* [135] and *C. ljungdahlii* [136], respectively, revealed that more ethanol was formed over acetate at high H_2_/CO ratio due to the design of gas distribution system that continuously supplied the gas into bioreactor, so the cell had adequate gas availability for further conversion of acetate to ethanol [136]. The gases of CO and H_2_ provide electrons and reducing equivalents to produce more reduced compounds rather than acetate [137]. Moreover, CO and hydrogen concentration must also be controlled for acetate production purpose.

The ratio of CO/CO_2_ is important to control when producing acetate from syngas. By increasing the CO/CO_2_ ratio, cell growth is increased, and the ratio of acetate/more electron-dense product will diminish [109]. The presence of more CO provides more electrons for reduction reactions, apart from H_2_. Acetate or acetyl Co-A that are initially formed will be reduced while being transformed into more electron-dense products causing lower acetate production. In addition, higher CO available in fermenters could inhibit some acetogens showing that CO concentration needs to be controlled.

### 4.3. Components of Fermentation Media

Essential WLP enzymes, including FDH, monofunctional CODH, bifunctional CODH/ACS, H_2_ase, PTA, and ACK, have specific cofactors involved in the catalytic reactions, which will influence acetate formation [138,139]. In acetogens, FDH has both iron (Fe), tungsten (W), selenium (Se), and/or molybdenum (Mo) [140] in its structure. The addition of these metals to fermentation media can, therefore, positively impact the FDH activity as their presence serves as cofactors mediating the reduction of CO_2_ to formic acid [138,141]. On the other hand, two other key enzymes of acetyl CoA accumulation, CODH, and ACS, have components such as Fe, nickel (Ni), and sulfur (S) metals clusters [114] in their active sides. Saxena et al. reported that more Ni in the fermentation media up to 8.4 µM could increase both the CODH activity of *C. ragsdalei* and result in increased acetate formation. Nickel removal from the media decreased cell growth; this indicated that Ni presence was highly necessary for CO fermentation to acetate. Ragsdale [124] also found that two major CODH classes involve Mo-Cu-S-CODH and Ni-CODH (monofunctional Ni-CODH and bifunctional CODH/ACS). The adjustment of the media containing such components could significantly affect both reversible catalytic oxidation of CO to CO_2_ by monofunctional CODH to provide reducing equivalents and acetyl CoA formation from CO by bifunctional CODH/ACS according to the WLP pathway (See Figure 2) [138].

Another essential enzyme, hydrogenase, contains Ni and Fe for producing reducing equivalents via hydrogen oxidation. The presence of Ni and Fe in hydrogenase has been described for *M. thermoacetica* [142] or *C. pasteurianum*, which only has Fe for hydrogen production via proton reduction [143]. Hence, the addition of both Ni and Fe into fermentation media could promote H_2_ase activity to release sufficient reducing equivalents used for acetate formation. The two last enzymes, PTA and ACK, required Mo for acetate production. The absence of Mo led to downregulated gene expression levels and is associated with the decrease of acetate formed [139]. Han [139] also reported that Ni, Fe, and Mo removal caused a diminishment of acetate concentration during syngas fermentation by *C. carboxidivorans* P7. Nevertheless, the medium without Se and Co could increase the formation of acetate by this bacterium. Kundiyana [117] also reported that acetate rose if the CoCl_2_ was present due to the transfer of the methyl group from THF to the cobalt center of CoFeSP. Another study revealed that increasing Ni^2+^ up to 8.4 µM and eliminating Fe^2+^ and WO^4−^ in the medium of *C. ragsdalei* could lead to higher acetate concentration [138]. On the other hand, the presence of Co^2+^, Cu^2+^, Mn^2+^, Mo^6+^, and SeO_4_^−^ in fermentation media had no influence on acetate production. According to these studies, we can conclude that each strain of acetogen requires specific metal micronutrients in specific amounts. Therefore, the use of metal cofactors must be adequately controlled to prevent that the gene expression is regulated towards biosynthesis of biomass instead of products. Accordingly, the optimization of metal components is required for maximum acetate production.

In addition to metal cofactors, other additional nutrients must be added in their optimal concentrations, such as yeast extract [137]. Yeast extract favored cell growth and had a positive impact on acetate production during syngas fermentation [72]. Nonetheless, yeast extract is expensive and needs to be replaced to improve the economy [144]. Several less-price materials for yeast extract substitutes are corn steep liquor (CSL), malt extract, or vegetable extract that has 14%, 29%, and 55% of the cost of yeast extract, respectively [145]. The CSL is generated from the corn wet-milling process and contains amino acids, minerals, and carbohydrates. The higher carbohydrate content is present in malt extract obtained from malt barley dehydration. Another yeast extract substitute, vegetable extract, is an excellent nitrogen source because this component has higher total nitrogen content and some minerals. Furthermore, hydrolyzed cotton seed flour, hydrolyzed soy flour, and ethanol stillage also can be the potential alternatives for yeast substitutes [146].

Vitamins are necessary for cell growth and targeted product accumulation. For instance, calcium pantothenate, the calcium salt of vitamin B5, is the acetyl-CoA precursor for acetate formation by homoacetogen [117]. Another essential vitamin for syngas-to-acetate fermentation is vitamin B12, an integral component of the cobalamin-dependent MeTr enzyme [147]. The addition of vitamin B12 could increase acetate concentration during syngas fermentation [117].

## 5. Crude Syngas Purification

### 5.1. Syngas Impurities

The undesirable gases or other toxic compounds in crude syngas formed during gasification adversely impact fermentation [38]. For example, tar consisting of oxygenates, phenolic compounds, olefins, and polyaromatic hydrocarbons [148], led to the longer cell adaptation phase, slow cell growth rate, and low acetate accumulation due to acetate redistribution to ethanol [38,117,149].

In addition to tar, nitrogenous species (NH_3_, HCN, and nitrogen oxides/NO_X_), another gasification contaminant, could also cause syngas fermentation interference if dissolved in large amounts into fermentation media. Ammonium (NH_4_^+^) is necessary as a nitrogen source for some acetogens and could enhance cell biomass and acetate production at certain concentrations [43]. The typical concentration of NH_3_ in real crude syngas (4500 ppm [150] corresponding to 7.1 g/L NH_4_Cl) insignificantly affects *C. carboxidivorans* growth, but inhibition of cell growth and less acetate accumulation occurred above 7.5 g/L NH_4_Cl [43]. Addition of 158 mol/m^3^ NH_4_^+^ (8.45 g/L NH_4_CL) into the fermentation media of *C. ragsdalei* reduced cell production and hydrogenase activity [151]. The hydrogenase provides reducing equivalents from hydrogen oxidation through Wood–Ljungdahl Pathway for the CO_2_ reduction into acetate. Insufficient reducing equivalents might cause less acetate formation. The NH_4_^+^ reacting with Cl^−^ from HCl, another syngas impurity, could also increase the osmolarity of the fermentation broth causing cell damage [37].

Another nitrogenous species that also inhibits hydrogenases is NO. The activity of this enzyme started being interfered with by the presence of NO above 40 ppm (0.072 µM) and decreased to a steady level at 160 ppm (0.288 µM) due to cell growth inhibition of *C. carboxidivorans* [152]. This study also reported that after NO exposure, product changes from acetate to ethanol occurred. It indicates that NO promoted undesired solventogenesis which led to a decrease in the acetate accumulation by *C. carboxidivorans*. The presence of NO_2_ also could interfere with the formate dehydrogenase (FDH) another essential enzyme for syngas fermentation [42].

Minor nitrogenous components of crude syngas, such as cyanide (CN^−^), act as a intracellular enzyme inhibitor of acetogens, particularly for CODH [153]. The CN ligand, a CO analog, reversibly binds the Ni ion in the C-cluster of CODH, and this binding is stabilized by hydrogen-bonding interactions between the nitrogen and histidine/lysine [115]. Cyanide can be categorized as a rapid or slow CODH binding inhibitor with different binding modes, such as multiple sites, bent, or linear binding [113]. Inhibition by CN caused a decreased growth rate and lower biomass concentration when cyanide concentration in syngas was up to 1.0 mM [153]. In addition, the culture required a longer time for adaptation if cyanide was present in the syngas.

Enzyme and cell growth inhibition also occurs when it is contaminated by sulfurous products, such as H_2_S, COS, and/or SO_2_ in the syngas is dissolved into the fermentation media. The acetogens can still tolerate a low concentration of H_2_S; for instance, *C. carboxidivorans* can grow well and rapidly consume CO and CO_2_ in the fermentation media containing 0.1–1 g/L H_2_S [43]. However, when H_2_S concentration was elevated to 2 g/L in cultures of *C. carboxidivorans*, the adaptation periods of the cells to this concentration were up to 132 h. Another sulfurous species, COS (carbonyl sulfide) was shown to result in CODH inhibition, thereby limiting the supply of reducing equivalents for CO_2_ reduction into acetate [42].

### 5.2. Syngas Cleanup Techniques

Crude syngas will need cleanup to remove unwanted components, prior to fermentation. Crude syngas purification generally uses two approaches, cold or hot gas cleanup [154]. Cold gas cleanup is the conventional technique performed at low temperatures (below ambient temperature), whereas hot gas cleanup employs higher temperatures (>300 °C). The use of low temperatures in cold gas cleanup impacts the need for energy consumption for cooling the gas as gasification is operated at 800–900 °C. Instead, hot gas cleanup can solve the efficiency loss problem and enable converting the waste into beneficial products.

Each impurity has specific recommended purification techniques. Purification of tar-contaminated syngas through cold gas cleanup approach can employ wet scrubbing with a liquid absorbent such as water or oil. The oil-based absorbent is preferred over water since this adsorbent has high efficiency in removing heavy and heterocyclic tar compounds [121] and can be operated at high temperatures while being regenerated by hot gas stripping [52]. In contrast, the use of water for tar removal generates more liquid waste that impacts the cost of wastewater treatment. Wet scrubbing using water can also be applied for nitrogenous species or chloride-contaminated syngas purification [136]. However, the caustic solution is more attractive for chloride removal because of the easy separation of precipitates formed from chloride and NaOH reaction. Caustic-based scrubbing could also eliminate COS as a toxic contaminant for syngas fermentation [155]. Meanwhile, cold gas cleanup for sulfurous species removal, apart from physical absorption, involved a chemical reaction [156]. Such absorbents include methyl ethanolamine (MEA), diglycolamine (DGA), diethanolamine (DEA), di-isopropanolamine (DIPA), methyl diethanolamine (MDEA), triethanolamine (TEA), potassium carbonate for chemical solvents, polyethylene glycil alkyl ethers, methanol, N-methyl-2-pyrrolidone, propylene carbonate for physical solvents. Tar removal using hot gas cleanup involves such as thermal cracking, hydrocracking, and steam or dry reforming [156]. Tar removed by thermal cracking produces C and H_2_, which also requires a high temperature above 1100 °C. In the hydrocracking of tar, cracking tar needs hydrogen, and methane is produced as the output. Lastly, reforming can transform tar into CO and H_2_. Using a proper catalyst can reduce the need for a temperature up to 650 °C with higher conversion.

The hot gas cleanup approach for the removal of nitrogenous species can cope with wet scrubbing drawbacks such as a high waste stream. The potential catalyst used includes alkaline metal catalysts (dolomite, olivine, CaO, and MgO) [157], iron- [158], nickel- [159], and ruthenium- [160] based catalysts. For example, ammonia, one of the major nitrogenous contaminants, can be decomposed through dehydrogenation into N_2_ and H_2_ catalyzed by calcined dolomite, nickel, and iron catalyst. This technique, however, has drawbacks in terms of the temperature use reaching above 850 ℃ and catalyst durability issues [161]. Another approach widely reported to remove ammonia is selective oxidation [162]. Ammonia reacts with an oxidizer such as O_2_, NO, or NO/O_2_ mixture to form N_2_ and H_2_O. Nevertheless, the presence of O_2_ and NO can further decrease H_2_ or CH_4_ concentration and form undesirable contaminants such as NO_2_. Thus, the selection of suitable catalysts acting on NH_3_ without interactions with H_2_ and CH_4_ is important for this process.

Desulfurization with hot gas cleanup employs a sorbent such as zeolites, activated carbon, and metal oxides with mesoporous silica, graphite oxide, and hydroxyapatite as the supports [156]. Metal oxide (MO_x_) is the common sorbent for sulfur removal and can be regenerated as a reaction below. Metal oxide catalysts are based on ZnO or CuO as the most common catalysts as well as alkaline earth metal-based sorbents (CaO, CaCO_3_, and dolomite [163,164]. The process of hot gas cleanup to remove HCl in the syngas is dehydrochlorination using metal catalysts (MO, MCO_3_, and M_2_CO_3_) [165]. Various types of metal catalysts are used for HCl separation, such as alkali metal (Li, Na, and K) [166], alkali earth metal (Mg, Ca, Sr, and Ba) [165], and transition metal oxide (MnO, CoO, ZnO, NiO, Y_2_O_3_, Cu_2_O, and FeO) [156].

## 6. Minimization of Mass Transfer Limitation

Gas/liquid mass transfer limitation becomes a critical issue when the fermenter size is increased and the volume of fermentation media to surface volume further increases. Gaseous substrates (CO, CO_2_, and H_2_) as part of syngas fermentation have low aqueous solubilities, even lower than oxygen solubility, and low absolute and partial pressure leading to the slow gas transfer rate in the liquid fermentation medium [167]. Modifying the bioreactor design and configuration and/or adding some chemical agents, such as surfactants, can solve this problem. Previous studies have determined the mass transfer rate-expressed by K_L_a values (volumetric mass-transfer coefficient) using both a continuous stirred tank reactor (CSTR), a bubble column reactor (BCR), an airlift reactor (ALR), and some modified bioreactors systems based on biofilm formation (Table 6).

### 6.1. Bioreactor Design

The continuous stirred tank reactor (CSTR) has been extensively used in the biotechnological industries due to its simple operation principle and good mixing abilities. However, in gas/syngas fermentation, the CSTR requires high agitation speed, long mixing duration, extended bioreactor pressure, an appropriate agitator, and a baffle design to increase the gas-liquid interfacial area solution for mass transfer issues [128,176]. The need for impeller speed acceleration, however, impacts high agitation power input per volume used for bubble breakup. Thus, the fermentation system requires high energy consumption associated with high costs. Furthermore, the high agitation speed can damage shear-sensitive bacteria that will interfere with the fermentation stability. Therefore, syngas fermentation using a CSTR is not a good option for larger-scale production [53]. The bubble column reactor (BCR) and airlift reactor (ALR) can be alternatives due to lower energy consumption and better gas/liquid mass transfer efficiency than the CSTR.

The basic design of the BCR is a liquid-filled cylindrical vessel with a gas sparger forming bubbles at the bottom of the reactor [51]. The gas bubbles convectively spread throughout the column volume and create mixing. In this reactor, the gas/liquid mass transfer efficiency can be increased by reducing the bubble diameter using microbubble dispersers (micro-sparger) and elevating the gas flow rate to provide a higher gas-liquid interfacial area [53,117]. Nevertheless, because good mixing will be reached at the high gas flow rate, the syngas will be incompletely converted in the exhaust line, so gas recirculation is required in this bioreactor [168]. On the other hand, another bioreactor, an airlift reactor (ALR) equipped with a draft tube, can assist gas circulation, and provide higher residence time for syngas fermentation. In addition, some modifications of the BCR, such as the BCR with cell immobilization [53] or with cell recycling and simultaneous gas feeding [54], can solve the problem of cell washout due to the high dilution rate and increase cell density and acetate production. The BCR has, however, been found to produce undesirable foaming issues demanding the use of antifoam additions.

Entrapping the biocatalyst cells on a solid surface is a useful strategy for extending the retention time of the cells in the bioreactor while ensuring a high cell density. As a result, cell growth-associated products such as acetate from syngas will produce higher concentrations per unit of a bioreactor with low washed-out of biocatalyst cells. This strategy can be conducted using a biofilm reactor where the microorganism is attached to a solid surface, such as the extracellular polymeric matrix (EPM), and where the gas is passed through the reactor. This bioreactor type can provide direct interaction between syngas and microorganisms and give a longer retention time for gas/nutrient uptake into the cell. Moreover, the scaffold containing synergistic microbial consortia is more stable and resistant to antimicrobials and other toxic compounds such as CO toxicity. Gunes [55] mentioned the classification of biofilm-based reactor systems, including the hollow-fiber membrane biofilm reactor (HFMBR), trickle bed reactor (TBR), rotating packed bed biofilm reactor (RPBR), monolithic biofilm reactor (MBR), and bulk-gas-to-atomized liquid (BGAL) reactor as described in Figure 4.

A hollow fiber membrane placed inside the syngas fermenter could assist the formation of a biofilm population on the outer microporous membrane surface. Therefore, this bioreactor is named a hollow fiber membrane biofilm reactor (HFMBR). In this bioreactor design, the syngas flows inside the hollow fiber module, from where it diffuses through the microporous membrane towards the liquid bulk region without forming bubbles (See Figure 4) [102]. Biofilm naturally grows and attaches to the outer membrane surface from where it grows by consuming the syngas. This bioreactor is designed to reduce energy consumption, shear stress, and reactor requirement over conventional bioreactors [56]. Orgill [57] also mentioned that this bioreactor has higher mass transfer characteristics than CSTR and BCR, relying on high agitation speed and high liquid flow rates, respectively. A common problem for this membrane-based process is fouling or biofouling due to excessive growth of the biofilm, which again lowers gas flux through the membrane. Thus, bioreactor improvement includes ways to reduce biofilm thickness which causes biofouling. The selection of membrane material that can prevent liquid permeation through the membrane and has high resistance to high internal gas pressure is further important. An example of this material is using a non-porous ultra-thin layer, which is hydrophobic [172].

The biofilm can also grow in a packed bed column receiving carbon and donor electron sources from the syngas and other nutrients from the liquid medium flowing through the trickle bed reactor (TBR). This reactor is generally equipped with an inlet spreader to provide good liquid nutrient and gas distribution and can be operated in either co-current or counter-current mode [55]. The TBR is simple to operate and maintain and can be operated under high pressure allowing for good transfer of gaseous materials into the liquid phase. Moreover, this reactor has a high fluid dynamic which is nearly the same as the plug flow regime [58]. Homogenous distribution of the liquid flowing down the bed surface, where the biofilm is highly concentrated, is the key to the success of this reactor use [55]. However, this reactor has several drawbacks involving the need for less viscous nutrients, foaming, maldistribution/channeling of liquid and gas on the bed, decreasing mass transfer rate, and flooding risk encountered in countercurrent flow. The flooding issue could be mitigated by altering the gas flow direction from countercurrent to co-current mode [59,71]. Devarapalli et al. reported that this reactor was usable for acetate production as well as alcohols [71]. Supplying syngas at double the rate from 2.3 to 4.6 standard cubic centimeters per minute (sccm) favored biofilm growth in the TBR and promoted acetate production instead of ethanol production.

Two other biofilm reactor types can potentially be used for syngas-to-acetate production: rotating packed bed biofilm reactor (RPBR) and a monolithic biofilm reactor (MBR). Shen [175] performed syngas fermentation by *C. carboxidivorans* using horizontal RPBR. In this bioreactor design, the cells are immobilized on the carrier inside a rotating cage attached to the axes of the bioreactor. Only half of the rotating cage is submerged, whereas the other half is contacted with bioreactor headspace (see Figure 4). Continuous rotation performed in this bioreactor allows the cells to have contact with the gas in the headspace and the nutrients in the liquid phase in sequences. In the reactor headspace, where the gas phase is the dominant region, diffusion from bulk gas to the gas-liquid interface can be eliminated due to direct contact between gas and the thin liquid film [61]. The gas is directly transferred across a thin liquid film covering the biofilm with lower transfer resistance, followed by transport to the cell. The other advantages of syngas fermentation using this reactor are less energy consumption due to the low rotation speed of 5 rpm and the operation at a wide range of gas pressure and flow rates compared to TBR, which requires a narrow range of flow rates to avoid flow channeling and flood problem [58,62]. This reactor also has a higher mass transfer efficiency value than a conventional disc-type rotating biological contactor (RBC) [63]. In addition, the KLa of CO in this bioreactor was lower than that in the CSTR, but the product accumulation could be higher due to the enhanced mass transfer when the rotating cage is in the headspace part [175].

The MBR (monolithic biofilm reactor) is the upgraded version of the BCR containing a bundle of uniformly narrow, straight, and parallel flow channels to minimize the cell washout during continuous operation mode [55]. The cells attach to the inner wall of the monolithic channel by consuming the gas flowing through this channel. Because the packing material of this bioreactor has a large open frontal area, the flow resistance, pressure drop, and energy loss can be minimized, which will influence the cost-effectiveness of a multiphase reaction system of a larger-scale MBR [177,178]. Even though this bioreactor type has a higher mass transfer efficiency than the BCR at certain conditions [168], the main problem is biofouling or clogging because of biofilm formation [177]. In addition to the aforementioned bioreactors, another bioreactor design, a bulk-gas-to-atomized-liquid reactor that provides higher energy efficiency and four times lower energy requirement than CSTR, has further been studied [48].

### 6.2. Addition of Mass Transfer Enhancers

The attachment of a microporous hollow fiber membrane to the bioreactor was found to produce bubble coalescence at the bubble generation site [179]. Bubble coalescence occurs when tiny bubble pairs that are released from membrane surface pores and suspended in the fermentation liquid collide and coalesce to generate bigger bubbles. Uncontrolled big bubble size distribution limits gas/liquid mass transfer and causes cell damage. Bubble coalescence could be prevented by adding inorganic electrolytes (salts), surfactants, or nanoparticles.

The addition of more electrolytes into fermentation broth, such as sulfate, nitrate, and chloride, has been found to counteract this problem and increase CO mass transfer [180]. Cations and anions from the ionization of salt in the fermentation medium form dense electrical double layers surrounding the gas bubble, causing repulsive force among the gas bubble, and no bubble coalesced [181]. Salt type and concentration affect mass transfer coefficient and bubble stabilization. Increasing salt concentration can stabilize the bubble and increase mass transfer coefficient, but high salt concentration in fermentation liquid can also inhibit the cell, so the optimization are needed [179].

In addition to the salt, the addition of the cationic surfactant such as hexyltrimethylammonium bromide (HTAB), octyltrimethylammonium bromide (OTAB), dodecyltrimethylammonium bromide (DTAB), and hexadecyltrimethylammonium bromide (HDTAB) could minimize the bubble coalescence and increase KLa [182]. The cationic surfactant is preferred for larger-scale use because of its more eco-friendly chemicals and less toxicity to microorganisms than anionic surfactants such as sodium dodecyl sulfate [183]. The cationic surfactant will surround the gas bubble with the inward orientation of hydrophobic tails and the outward orientation of hydrophilic heads. The positively charged bubbles will be formed and inhibit bubble coalescence by electro-repulsive force, so the gas-liquid interfacial area and gas/liquid mass transfer also will increase. In addition, the addition of surfactant also could alleviate cell agglomeration, causing an increase in gas consumption and product formation [127].

Adding nanoparticles into a liquid medium could increase the mass transfer rate. The enhancement of the mass transfer rate could be described with three proposed mechanisms reported by Kluytmans et al. [184]. First, the nanoparticle aims to facilitate gas bubble transport via an adsorption mechanism where the gas is adsorbed to a gas-liquid interfacial layer and then desorption to liquid bulk. The first mechanism is called the as shuttle or grazing effect. Second, the collision and interaction between particles and the gas-liquid interface could lower the effective diffusion layer, also called the hydrodynamic effect. Finally, the specific gas-liquid interfacial area (a) possibly increases due to the adherence of nanoparticles on the bubble surface to avoid bubble aggregation and expansion. The increase of gas-liquid interfacial area increases the K_L_a value as an indicator of mass transfer enhancement. The first and second mechanisms are highly affected by the specific characteristics of the nanoparticles, such as hydrophobicity. The hydrophobic nanoparticle could decrease the water layer thickness of the nanoparticle surface and increase the affinity between the particle and gas-liquid interface, causing more collision between them, and as a consequence of these, the K_L_a value could increase [185]. Several modified nanoparticles have been evaluated for syngas fermentation, such as the MCM41 nanoparticle [186], with mercaptopropyl functional groups [187], the methyl-functionalized silica nanoparticle [185], and methyl-functionalized mesoporous silica SBA-15 [188]. However, this nanocellulose is difficult to recover and reuse and comes with a high cost. Kim et al. [189] proposed the use of magnetic nanoparticles for easier recovery using methyl-functionalized silica and methyl-functionalized cobalt ferrite-silica nanoparticles.

## 7. Future Perspective of Acetate Production from Syngas

The integration of lignocellulose gasification and syngas fermentation is a promising technology for future acetate production as:Current commercial acetate production relies on petrochemical materials through methanol carbonylation, acetaldehyde oxidation, and the hydrocarbon oxidation process [10]. Therefore, the production leads to greenhouse gas emissions into the atmosphere. Lignocellulose feedstocks, on the other hand, are a renewable source without any current use. The use of lignocellulosic materials supports the decarbonization and recycling of waste materials while decreasing greenhouse gas emissions.Biomass gasification is a well-known technology producing syngas, which can be used directly for electricity production or converted biologically or by catalysis into biofuels and chemicals. Gasification makes use of the whole biomass material and, thereby, mitigates the problem of left over lignin products using the biomass-to-sugar route.Even though chemical routes of acetate production from CO through methanol carbonylation have been intensively studied and are commercially used on an industrial scale, this process still requires extensive purification of the syngas, increasing the cost of the overall process. High cost and low reaction times is further an issue for the chemical routes based on catalysis. Biological syngas fermentations can, therefore, be the alternative to make use of syngas in the future.

Homoacetogens, biocatalysts producing acetate as a sole product from syngas, are promising microorganisms for acetate production over the other types of autotrophic acetogens. Furthermore, instead of mesophilic homoacetogens, the use of thermophilic homoacetogens such as *M. thermoacetica* brings about further advantages in larger-scale applications due to high acetate productivity, titer, acetate formation rate, gas utilization efficiencies, and less contamination. The research on syngas fermentation by *M. thermoacetica* using simulated syngas without impurities has shown a maximum titer and productivity of 31 g/L and 0.55 g/(L.h), respectively [90]. However, this research still used simulated syngas without the presence of impurities. A high concentration of impurities in a gaseous mixture of real syngas inhibits cell growth and acetate formation, as discussed in this review. Therefore, impurities removal is required to optimize syngas fermentation for acetate production.

Techno-economic evaluation of lignocellulose-derived syngas production reported by He et al. and Regis et al. showed that the major contribution to the production cost was the syngas cleanup stage, up to 25–39% of the total capital cost depending on the lignocellulose type [190,191]. The use of a robust microorganism capable of utilizing syngas from various types of biomass feedstock and which has resistance towards syngas impurities and differences in composition can reduce the syngas purification cost. Such a fermentation process can be improved by genetic and metabolic engineering to obtain a more effective microorganism. Moreover, the fermentation conditions can be optimized. The complete genome sequence of *M. thermoacetica* has been reported [142], easing research on ways to improve this microorganism. Until now, studies have focused on engineered *M. thermoacetica* for the production of non-native products such as acetone [192]. Studies on genetic modification of *M. thermoacetica* for improving tolerance towards syngas impurities while producing acetate are needed in the future.

## 8. Conclusions

In summary, acetate production through the integrated process of lignocellulose gasification and syngas fermentation offers a commercially interesting solution to the challenges of petrochemical-derived or sugar-derived acetate production and meets demands of the circular bioeconomy. The full use of the entire lignocellulose feedstocks or waste material is a major benefit of this process, along with the limited need for syngas purification compared to catalytic processes. Finally, the relatively low need for energy input makes this process environmentally and economically attractive. The use of acetate-producing thermophilic homoacetogens such as *M. thermoacetica* is industrially preferable for syngas-to-acetate conversion due to high selectivity, yield, and productivity over other autotrophic acetogens and mixed cultures that generally produce lower acetate yield as reviewed in this article. Finding a robust bacterium that can resolve the problems of inhibition by gasification-generated syngas impurities and CO toxicity while possessing high acetate yield and productivity is still a target for improvement of the known microbes by metabolic and genetic engineering research. Finally, further research should focus on improving bioreactor design to enhance the mass transfer of syngas into the fermentation broth.

## Figures and Tables

**Figure 1 microorganisms-11-00995-f001:**
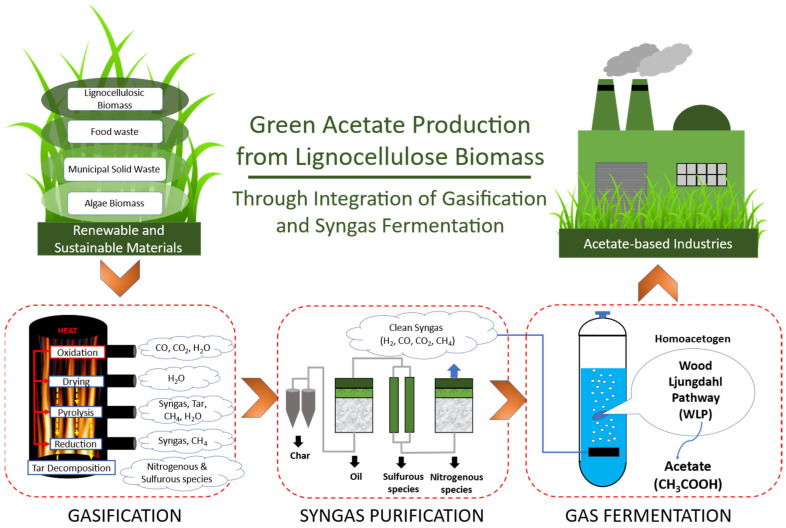
Acetate production through gasification of lignocellulosic biomass materials followed by syngas fermentation.

**Figure 2 microorganisms-11-00995-f002:**
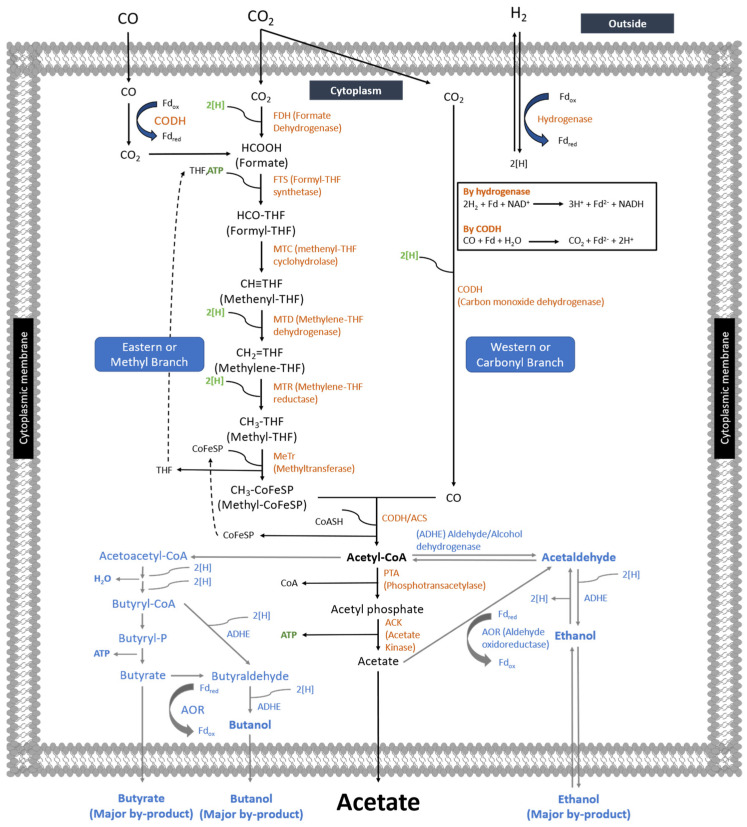
Wood–Ljungdahl Pathway of *C. carboxidivorans* using C1 gases (CO and CO_2_) [49].

**Figure 3 microorganisms-11-00995-f003:**
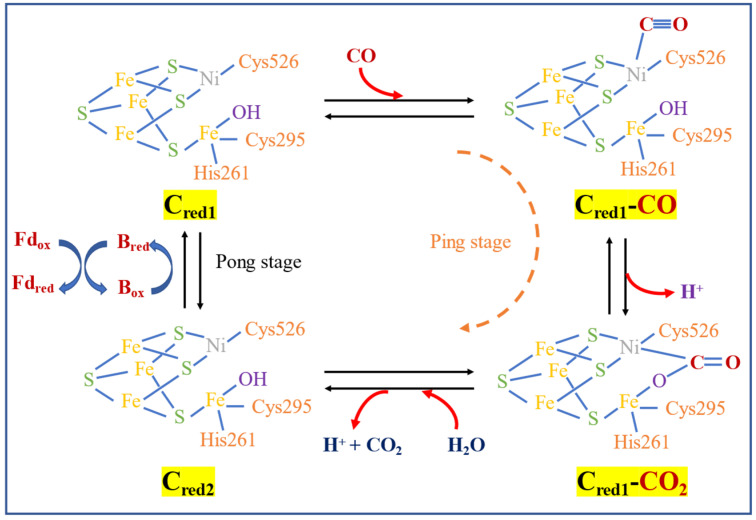
The ping pong reaction of CODH [113].

**Figure 4 microorganisms-11-00995-f004:**
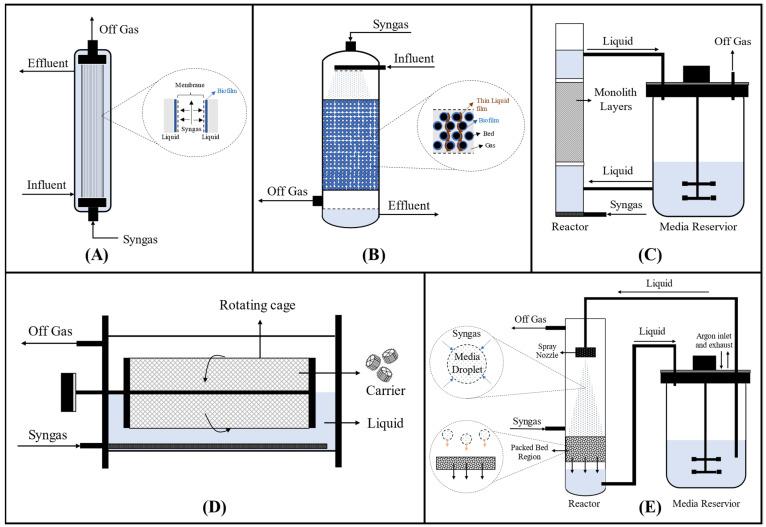
Schematic description of biofilm reactors: (**A**) Hollow-fiber membrane biofilm reactor (**B**) Trickle bed reactor (**C**) Monolithic biofilm reactor (**D**) Rotating packed bed biofilm reactor (**E**) Bulk-gas-to-atomized-liquid reactor.

**Table 1 microorganisms-11-00995-t001:** Carboxydotrophic microorganisms.

Microorganism	Opt. Temp. [°C]	Opt. pH [-]	Other Dominant Products	Ref.
**Mesophilic bacteria**				
*Clostridium ljungdahlii*	37	6–7	Ethanol, lactate, 2,3-butanediol	[47,48]
*Clostridium carboxidivorans*	37	6	Ethanol, butyrate, butanol, hexanoate, hexanol	[37,49]
*Clostridium ragsdalei*	37	6	Ethanol, lactate, 2,3-butanediol	[50,51]
*Clostridium autoethanogenum*	37	6.5	Ethanol, lactate, 2,3-butanediol	[52,53]
*Clostridium aceticum*	30	8–8.5	-	[54,55]
*Clostridium drakei*	25–30	5.8–6.9	Ethanol, butyrate	[56]
*Acetobacterium woodii*	30	7.0	-	[57]
*Peptostreptococcus productus*	37	7.0	-	[58,59]
*Alkalibaculum bacchii*	37	8.0–8.8	Ethanol	[60]
*Butyribacterium methylotrophicum*	37	7.0	Ethanol, butyrate, butanol	[61]
*Eubacterium limosum*	37	7.0	-	[62]
*Oxobacter pfennigii*	36–38	7.3	Butyrate	[63]
**Thermophilic bacteria**				
*Moorella thermoacetica*	55	6.5–6.8	-	[46]
*Desulfotomaculum kuznetsovii*	60	7.0	H_2_S	[64,65]
*Desulfotomaculum thermobenzoicum* subsp. *thermosyntrophicum*	55	7.0	H_2_S	[65,66]
**Mesophilic archaea**				
*Methanosarcina acetivorans*	37	7	CH_4_, formate	[42]
**Thermophilic archaea**				
*Archaeoglobus fulgidus*	83	6.4	Formate, H_2_S	[67,68]

**Table 2 microorganisms-11-00995-t002:** The potential reaction of acetic acid, ethanol, butyric acid, butanol, hexanoic acid, and hexanol [67,68].

Product	Reactions	∆G° [kJ/mol]
Acetic Acid	4 CO + 2 H_2_O → CH_3_COOH + 2 CO_2_	−154.6
	4 H_2_ + 2 CO_2_ → CH_3_COOH + 2 H_2_O	−74.3
	2 CO + 2 H_2_ → CH_3_COOH	−114.5
Ethanol	6 CO + 3 H_2_O → C_2_H_5_OH + 4 CO_2_	−217.4
	6 H_2_ + 2 CO_2_ → C_2_H_5_OH + 3 H_2_O	−97.0
	2 CO + 4 H_2_ → C_2_H_5_OH + H_2_	−137.1
	3 CO + 3 H_2_ → C_2_H_5_OH + CO_2_	−157.2
Butyric Acid	10 CO + 4 H_2_O → CH_3_(CH_2_)_2_COOH + 6 CO_2_	−420.8
	10 H_2_ + 4 CO_2_ → CH_3_(CH_2_)_2_COOH + 6 H_2_O	−220.2
	6 CO + 4 H_2_ → CH_3_(CH_2_)_2_COOH + 2 CO_2_	−317
Butanol	12 CO + 5 H_2_O → C_4_H_9_OH + 8 CO_2_	−486.4
	12 H_2_ + 4 CO_2_ → C_4_H_9_OH + 7 H_2_O	−245.6
	6 CO + 6 H_2_ → C_4_H_9_OH + 2 CO_2_ + H_2_O	−373
	4 CO + 8 H_2_ → C_4_H_9_OH + 3 H_2_O	−334
Hexanoic Acid	16 CO + 6 H_2_O → CH_3_(CH_2_)_4_COOH + 10 CO_2_	−656
	16 H_2_ + 6 CO_2_ → CH_3_(CH_2_)_4_COOH + 10 H_2_O	−341
	16 CO + 6 H_2_O → CH_3_(CH_2_)_4_COOH + 4 CO_2_	540
Hexanol	18 CO + 7 H_2_O → C_6_H_13_OH + 12 CO_2_	−753
	18H_2_ + 6 CO_2_ → C_6_H_13_OH + 11 H_2_O	−395
	6 CO + 12 H_2_ → C_6_H_13_OH + 5 H_2_O	−514

**Table 4 microorganisms-11-00995-t004:** Acetate production from syngas by mixed culture inoculum.

Inoculum Source	Dominant Microorganism	Temp. [°C]	pH [-]	Acetate Concentration [g/L]	Ref.
Sludge from a starch-containing biogas reactor	*Thermoanaerobacterium* and *Thermohydrogenium*	55	6.0	42.4	[102]
Mesophilic methane production reactor containing glucose and acetate	*Clostridium* spp. (*C. ljungdahlii* and *C. drakei*)	35	4.5–4.8	12.5	[104]
Mesophilic USAB reactor for cassava stillage waste treatment	Not applicable	37	6.5–7.5	1.3	[105]
Sewage sludge from a sewage treatment plant	*Clostridium* and *Acetobacterium*	37	9.0	3.1	[103]
Waste activated sludge	*Natronincola* and *Desulfitispora*	37	9.0	8.1	[60]

**Table 5 microorganisms-11-00995-t005:** Detailed metabolic reactions of Wood–Ljungdahl Pathway (WLP).

Eq.	Reaction	Enzymes	Ref.
The Eastern Branch of WLP			
(5)	NADPH + CO_2_ + H^+^ ↔ NADP^+^ + HCOOH (formate)	Formate dehydrogenase (FDH) (EC 1.2.1.43)	[44]
(6)	HCOOH (formate) + ATP + H4folate ↔ 10-HCO-H4folate (formyl-THF) + ADP + Pi	Formyl-THF synthetase (FTS) (EC 6.3.4.3)	
(7)	10-HCO-H4folate (formyl THF) + H+ ↔ 5,10-methenyl-H4folate (methenyl-THF)	Methenyl-THF cyclohydrolase (MTC) (EC 3.5.4.9)	
(8)	NAD(P)H + 5,10-methenyl-H4folate (methenyl-THF) ↔ 5,10-methylene-H4folate (methylene-THF) + NAD(P)	Methylene-THF dehydrogenase (MTD) (EC 1.5.1.5, NADP; EC 1.5.1.15, NAD)	
(9)	2e^−^ + H^+^ + 5,10-methylene-H4folate ↔ 5-methyl-H_4_folate (methyl-THF)	Methylene-THF reductase (MTR) (EC 1.1.99.15)	
**The Western Branch of WLP**			
(10)	CoFeSP (Corrinoid iron-sulfur protein) + 5-methyl-H_4_folate (methyl-THF)↔ methyl-CoFeSP + H_4_folate (THF)	Methyl-H4folate:CoFeSP methyltransferase (MeTR) (EC 2.1.1.X)	[111,112]
(11)	CO_2_ + 2H^+^ + 2e^−^ ↔ CO + H_2_O	Monofunctional CO dehydrogenase (CODH) (EC 1.2.7.4)	
(12)	Methyl-CoFeSP + CO + CoASH → CH_3_COSCoA (Acetyl-CoA) + CoFeSP	Bifunctional CO dehydrogenase (CODH (EC 1.2.7.4)/Acetyl CoA synthase (ACS) (EC 6.2.1.1)	
**Acetyl CoA-to-Acetate Pathway**			
(13)	CH_3_COSCoA (Acetyl-CoA) + HPO_4_^2−^ → CH_3_CO_2_PO_3_^2−^ (Acetyl phosphate) + CoASH	Phosphotransacetylase (PTA) (EC 2.3.1.8)	[47]
(14)	CH_3_CO_2_PO_3_^2−^ (Acetyl phosphate) + ADP → CH_3_CO_2_PO_3_^2−^ (Acetate) + ATP + H^+^	Acetate Kinase (ACK) (EC 2.7.2.1)	

**Table 6 microorganisms-11-00995-t006:** Mass transfer coefficient of syngas for various types of bioreactor design.

Bioreactor	Microorganism	Gas	Optimum KLa [h^−1^]	Ref.
Continuous Stirred Tank Bioreactor	Not applicable	CO	155	[128]
Bubble Column Bioreactor	*C. carboxidivorans*	CO	400	[168]
Airlift combined with a 20-µm bulb diffuser	Not applicable	CO	91	[133]
Airlift reactor combined with single-point gas entry	Not applicable	CO	45	[133]
Hollow fiber membrane bioreactor	*Eubacterium limosum*	CO	155	[169]
Hollow fiber membrane bioreactor	Not applicable	H_2_ CO	840 420	[170]
External microporous hollow fiber membrane bioreactor	Not applicable	CO	385	[171]
Composite hollow fiber (CHF) membrane bioreactor	Not applicable	CO	947	[172]
Ceramic membrane bioreactor	Not applicable	CO	114	[173]
Trickle bed reactor	Sulfate-reducing bacteria	CO H_2_	121 335	[174]
Trickle bed reactor	*C. ljungdahlii*	CO	137	[174]
Horizontal rotating packed bed biofilm reactor	Not applicable	CO	73	[175]
Monolithic biofilm reactor	*C. carboxidivorans*	CO	450	[168]
Bulk-gas-to-atomized-liquid reactor	*C. carboxidivorans*	CO	137	[48]
Continuous Stirred Tank Bioreactor	Not applicable	CO	155	[128]
Bubble Column Bioreactor	*C. carboxidivorans*	CO	400	[168]

## Data Availability

Not applicable.

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
