# Peer review of "Acetate Production from Syngas Produced from Lignocellulosic Biomass Materials along with Gaseous Fermentation of the Syngas: A Review"

_microorganisms, 2023, doi:10.3390/microorganisms11040995_

Round 1

Reviewer 1 Report

The manuscript by Harahap and Ahring (Acetate Production from Syngas Produced from Lignocellulosic Biomass Materials along with Gaseous Fermentation of Syngas: A Review) is certainly a Review within the scope of Microorganisms. In general the manuscript is well written and brings new information (as a review) regarding the proposed topic, including nice figures, tables, microorganisms and enzymatic reactions involved. However, some improvements are needed before the manuscript is accepted for publication:

In L81-82 the authors claim that “no review article has discussed syngas-to-acetate fermentation”..... however, this reviewer found several reviews dealing with that, not cited by the authors. Authors should include the following reviews, highlighting their contributions and/or limitations, and how the present review moves the area forward:

- Sci. Total Environ. 840: 156663, 2022.

- Chemosphere 299: 134425, 2022.

- Front. Microbiol. 13: 865168, 2022.

- Biotechnol. Adv. 58: 107886, 2022.

- Appl. Microbiol. Biotechnol. 103: 8689-8709, 2019.

- Curr. Opin. Chem. Biol. 41: 84-92, 2017.

L83 should be “done in the present review paper”

L133-134 needs revision (“acetate accumulated in the fermentation broth from where it needs to be separated”?)… I believe “from where it” should be removed.

L221 should be “Savage et al

L259-269 should make reference to Fig. 2.

L331-332, the enzyme’s abbreviated names should be spelled in full, and make reference to the corresponding reactions shown in Table 4. Same for L345. (or authors should add the enzyme abbreviations in Table 4).

L365 should be revised “high-affinity towards cell membrane”?

L372-373 needs revision. How pH will “Lower (the) numbers of CODH genes”?? Is the expression of the genes that change with pH?

Author Response

Dear Reviewer 1,

Thank you for your feedback and your comments. Please see the attachment of our response to your comments.

Thank you

Reviewer 2 Report

Dear Authors,

The paper is well written, but I still miss some depth in some places.

Chapter 1-3: I miss further information. E.g. the possible yield when using different bacteria.

ll. 151-152: You describe that it is easily controllable. Is this the case for all substrates? For all hydraulic residence times, all temperatures, and all organic loading rates? This seems to be a very general statement

Table 2: I can't really understand why this table is important here

LL. 199-217 A table comparing such data would be very helpful.

LL. 239-245: What kind of microorganisms are included in the "mixed culture"?

LL. 359-360: What is the decline?

LL 383-385: How can you do that without killing every other bacteria?

LL. 411:415: What temperature do they use? Is it a good idea to use psychrophilic conditions by checking the conversion rate?

Table 5: please explain KLa

LL.816: How you write kLA is changed here

Chapter 7: I miss a discussion of price and whether the purity is sufficient for economic use of syngas.

Author Response

Dear Reviewer 2,

Thank you for your feedback and your comments. Please see the attachment of our response to your comments.

Thank you

Reviewer 3 Report

This is a well written review summarizing the acetate production from lignocellulose. I did not spot any weak point and I believe it will  be a great addition to the literature.

Author Response

Dear Reviewer 3,

Thank you for your comments to our manuscript. 

Thank you

Round 2

Reviewer 1 Report

The authors have addressed all the points raised by this reviewer, and now the manuscript can be accepted for publication. However, there is still a last correction that is needed:

L398 "the lower CODH gene copy number" should be changed to “a lower CODH gene expression”…. it is a matter of gene expression, not gene copy number!